# Comparative Prevalence of Ineffective Esophageal Motility: Impact of Chicago v4.0 vs. v3.0 Criteria

**DOI:** 10.3390/medicina60091469

**Published:** 2024-09-08

**Authors:** Teodora Surdea-Blaga, Stefan-Lucian Popa, Cristina Maria Sabo, Radu Alexandru Fărcaş, Liliana David, Abdulrahman Ismaiel, Dan Lucian Dumitrascu, Simona Grad, Daniel Corneliu Leucuta

**Affiliations:** 12nd Department of Internal Medicine, Emergency County Hospital, 400003 Cluj-Napoca, Romania; dora_blaga@yahoo.com (T.S.-B.); cristina.marica90@yahoo.com (C.M.S.); lilidavid2007@yahoo.com (L.D.); abdulrahman.ismaiel@yahoo.com (A.I.); dan_dumitrascu@yahoo.de (D.L.D.); costinsimona_m@yahoo.com (S.G.); 2Faculty of Medicine, “Iuliu Hatieganu” University of Medicine and Pharmacy, 400012 Cluj-Napoca, Romania; radufr@gmail.com; 3Department of Medical Informatics and Biostatistics, “Iuliu Hatieganu” University of Medicine and Pharmacy, 400349 Cluj-Napoca, Romania; danny.ldc@gmail.com

**Keywords:** high-resolution esophageal manometry, ineffective esophageal motility, dysphagia, esophagogastric junction, ineffective swallow

## Abstract

*Background and Objectives*: The threshold for ineffective esophageal motility (IEM) diagnosis was changed in Chicago v4.0. Our aim was to determine IEM prevalence using the new criteria and the differences between patients with definite IEM versus “inconclusive diagnosis”. *Materials and Methods*: We retrospectively selected IEM and fragmented peristalsis (FP) patients from the high-resolution esophageal manometries (HREMs) database. Clinical, demographic data and manometric parameters were recorded. *Results*: Of 348 HREMs analyzed using Chicago v3.0, 12.3% of patients had IEM and 0.86% had FP. Using Chicago v4.0, 8.9% of patients had IEM (IEM-4 group). We compared them with the remaining 16 with an inconclusive diagnosis of IEM (borderline group). Dysphagia (77% vs. 44%, Z-test = 2.3, *p* = 0.02) and weight loss were more commonly observed in IEM-4 compared to the borderline group. The reflux symptoms were more prevalent in the borderline group (87.5% vs. 70.9%, *p* = 0.2). Type 2 or 3 esophagogastric junction morphology was more prevalent in the borderline group (81.2%) vs. 64.5% in IEM-4 (*p* = 0.23). Distal contractile integral (DCI) was lower in IEM-4 vs. the borderline group, and resting lower esophageal sphincter (LES) pressure and mean integrated relaxation pressure (IRP) were similar. The number of ineffective swallows and failed swallows was higher in IEM-4 compared to the borderline group. *Conclusions*: Using Chicago v4.0, less than 10% of patients had a definite diagnosis of IEM. The dominant symptom was dysphagia. Only DCI and the number of failed and inefficient swallows were different between definite IEM patients and borderline cases.

## 1. Introduction

Esophageal motility disorders and the diagnosis criteria used to define each motility disorder were modified for the fourth time since the implementation in everyday practice of high-resolution esophageal manometry (HREM). These changes were based on the knowledge that was gathered in the last years from clinical applications and research investigations involving HREM in esophageal motility disorders. Normal values for provocative tests were published [1]. Data about the use of supportive testing are currently available in order to inform clinical decisions in case of borderline situations [2]. The evaluation of swallows during sitting position and the use of solid swallows were also included in this new classification, particularly in borderline cases [3]. Based on the Chicago v4.0, there are two main categories of esophageal motility disorders: disorders of esophagogastric junction (EGJ) outflow and disorders of peristalsis [1]. Ineffective esophageal motility is identified using HREM by a high number of swallows followed by ineffective contractions, with a distal contractile integral (DCI) lower than 450 mmHg·s·cm, indicating weak contractions, or less than 100 mmHg·s·cm, indicating failed contractions [1]. Ineffective esophageal motility is considered the most frequent motility disorder, being reported in 20–30% of esophageal manometries [4] and in 50% of patients with dysphagia [5]. It is frequently observed in patients with gastroesophageal reflux disease (GERD) [6], collagen diseases [7], or diabetes mellitus [8].

The diagnosis criteria for ineffective esophageal motility (IEM) were significantly changed in the last version of Chicago 4.0 classification, with more stringent thresholds. Based on Chicago v4.0 a conclusive diagnosis of IEM is based either on at least 80% of ineffective swallows or at least 50% swallows with failed peristalsis. Another change comes from reclassifying fragmented swallows as ineffective. Therefore, fragmented peristalsis (FP) is no longer a distinct motility disorder of peristalsis. All these changes were meant to increase the number of conclusive diagnoses of motility disorders [1]. Patients presenting with 50 to 70% ineffective swallows during esophageal manometry are considered borderline, and the manometric findings are deemed “inconclusive for a diagnosis of IEM”. In such cases, further diagnostic evaluations are warranted to clarify the diagnosis. Additional testing may include combined impedance/HREM [9,10], which can provide insights into bolus transit and the correlation with symptoms. An impaired bolus clearance detected using impedance was directly associated with the percentage of ineffective swallows recorded during manometry. In addition, patients with IEM showed a marked decrease in impedance values, indicating poor esophageal clearance [9,10]. Barium esophagogram can visually assess bolus passage through the esophagus and identify areas of stasis, impaired transit, or delayed esophageal emptying [11]. Moreover, performing multiple rapid swallows (MRS) during high-resolution manometry can evaluate the esophageal contractile reserve; a lack of contraction reserve on MRS may support the diagnosis of IEM when the percentage of ineffective swallows is borderline [12].

For almost all the motility disorders, it was stated that manometric changes should be accompanied by clinically relevant symptoms, like dysphagia or noncardiac chest pain [1]. However, no relevant symptoms for IEM diagnosis were mentioned in Chicago v4.0, probably because the studies published thus far reported no correlation between the manometric changes and symptoms [13].

The primary objective of this study was to determine how the Chicago v4.0 criteria changed the IEM prevalence in a cohort of patients referred for esophageal manometry. The secondary objective was to describe the clinical and manometric characteristics of patients with IEM based on Chicago v4.0 compared to those that now fall in the category of “inconclusive diagnosis of IEM”, hence, leading to a better understanding of the implications of this new diagnostic distinction.

## 2. Materials and Methods

### 2.1. Study Design

From our database of esophageal HRM recordings (performed from November 2015 to April 2023), we selected all patients with a diagnosis of IEM or FP, based on Chicago v3.0 criteria. All identified tracings were reanalyzed and reclassified using the Chicago v4.0 criteria to assess how the prevalence of IEM was modified using the new criteria. Some of the patients were borderline based on the new classification system (“inconclusive for a diagnosis of IEM”). For the secondary objective of the study, we compared the clinical and manometric parameters between patients who were still classified as IEM based on Chicago v4.0 and the patients with an inconclusive diagnosis for IEM using Chicago v4.0 criteria.

### 2.2. Selection of Patients

This study was of a retrospective design. All tracings of patients diagnosed with IEM or FP based on Chicago v3.0, irrespective of their symptoms, were selected from our database of esophageal HRM recordings. Patients referred to the motility department were patients with unexplained dysphagia, chest pain, globus, or ENT symptoms, with normal findings in upper GI endoscopy, and patients with reflux symptoms without esophageal esophagitis that required manometry to determine the location of the LES prior to 24 h pH-impedance monitoring. All patients were instructed during their appointment to avoid drugs that can affect esophageal motility (prokinetics, calcium channel blockers, nitrates, opiates, and anticholinergics) at least 48 h before manometry. Patients with previous surgery/procedures at the level of the esophagogastric junction were excluded. All the manometries were performed with the patients in the supine position at 30°. Tracings with at least 7 correct 5 mL water swallows were included in the analysis. All manometries were performed by the same specialized nurse, using the same manometry system (Isolab Manometry System, Standard Instruments GmbH, Germany). The system records the pressures with a solid-state catheter, with 36 sensors (Unisensor^®^, Unisensor AG, Switzerland).

### 2.3. Manometry Parameters and Classification of the Swallows

Based on the Chicago v3.0, ineffective swallows were swallows classified as weak (with a DCI between 100 and 450 mmHg·s·cm) or failed (DCI < 100 mmHg·s·cm). Premature contractions (with a distal latency (DL) < 4.5 s) and a DCI < 450 mmHg·s·cm were also considered failed contractions. Swallows with large breaks (>5 cm) in the 20 mmHg isobaric contour and a normal DCI (>450 mmHg·s·cm) were classified as fragmented. According to Chicago v3.0, patients were classified as IEM if at least 50% of the swallows were ineffective. Patients with ≥50% fragmented contractions who did not meet the IEM criteria were classified as FP [14].

All tracings classified as IEM or PF using Chicago v3.0 were reanalyzed using the Chicago v4.0 criteria. A diagnosis of IEM was based either on the presence of more than 70% ineffective swallows (including weak, failed, or fragmented swallows) or at least 50% failed swallows. Tracings with 50% to 70% ineffective swallows, were considered inconclusive for IEM, based on Chicago v4.0 criteria [1].

The following manometric parameters were analyzed: EGJ morphology [15] and length, LES resting pressure, integrated relaxation pressure (IRP—the upper limit of the normal was set at 28 mmHg, based on Chicago v3.0 [15]), and the mean value of distal contractile integral (DCI). The esophageal length was measured between the lower border of the upper esophageal sphincter and the upper border of the LES. The EGJ length was measured between the upper limit of the LES and the lower limit of the LES or the lower limit of crural diaphragm (if LES and crural diaphragm were separated). Regarding esophageal peristalsis, we recorded the number of normal, hypotensive, failed, spastic, fragmented, or hypertensive contractions for each patient.

### 2.4. Clinical Data

The demographic and clinical data were collected from a standardized questionnaire filled by a specialized nurse before each manometry. We collected data on age, gender, height, weight, esophageal symptoms (dysphagia, chest pain, and reflux symptoms), ENT symptoms, cough, and belching. We recorded the presence of dyspeptic symptoms (like nausea, bloating, epigastric pain, satiety, and postprandial fullness) and the duration of dysphagia. The questionnaire included data about having undergone upper GI endoscopy and relevant previous procedures such as anti-reflux surgery, surgery for hiatal hernia, or endoscopic procedures (i.e., endoscopic pneumatic dilation) at the level of EGJ.

### 2.5. Statistical Analysis

The study population characteristics were expressed as number and percentages for categorical variables. To assess whether the variables followed a parametric or nonparametric distribution, we evaluated skewness and kurtosis values. Variables with skewness or kurtosis outside the range of −1 to +1 were considered non-normally distributed and were reported as median and interquartile range (25th and 75th percentiles). Parametric data were reported as mean ± SD. We compared differences in several patient characteristics between the patients with IEM based on Chicago v4.0 (IEM-4 group) and the patients with inconclusive diagnosis of IEM (borderline group) using the chi-square test, one-way analysis of variance (ANOVA), and Mann–Whitney depending on the normal or abnormal distribution of data. A *p* value < 0.05 was considered statistically significant. The statistical analysis was performed using Medcalc statistical software (version 20.305) (MedCalc Software Ltd, Ostend, Belgium).

The study protocol was approved by the Committee of Ethics of the University of Medicine and Pharmacy, Cluj-Napoca, Romania.

## 3. Results

Of 367 manometric recordings, 19 were failures. Therefore, the prevalence was based on the remaining 348 tracings that were analyzed. Based on Chicago v3.0, 43 patients (12.3%) had IEM and 3 patients (0.86%) had FP. Using Chicago v4.0 classification, we identified 31 patients (8.9%) with IEM. This was the IEM-4 group: 28 patients had IEM based on Chicago v3.0, 2 had Chicago 3.0 FP, and 1 patient remained “unclassified” based on Chicago v3.0 due to borderline changes between IEM and FP. We compared these 31 patients with the remaining 16 patients whose diagnoses, based on the Chicago v3.0 (15 with IEM and 1 with FP), were changed to inconclusive under the Chicago v4.0 criteria. 

Considering the manometries of patients without an obstruction at the EGJ or previous intervention on EGJ (143 patients), the prevalence of IEM based on Chicago v3.0 was 30.1% and, based on Chicago v4.0 IEM, the prevalence was 21.7%. A total of 17 patients (4.8% from 348 tracings or 11.8% from the 143 patients mentioned above) were reclassified using Chicago v4.0 (Figure 1).

All the patients included in this study, irrespective of their symptoms, had normal esophageal mucosa on upper GI endoscopy. The patients with reflux symptoms referred to manometry prior to ambulatory pH impedance had no esophagitis or Barret’s esophagus on prior endoscopy. They were evaluated for persisting reflux symptoms despite antisecretory treatment. Manometry was performed to determine the LES position before placement of the pH-impedance catheter.

### 3.1. Demographic Parameters and Symptoms

Patients in both groups had a similar age and BMI. The proportion of lean subjects (BMI < 25 kg/m^2^) was similar in both groups. Three patients had missing BMI data. Regarding the symptoms, only dysphagia was more commonly observed in the IEM-4 group, being reported by 77% of patients compared to 44% in the borderline group. Chest pain and weight loss were more frequent in the IEM-4 group but did not reach statistical significance. Reflux symptoms were more common in the borderline group but without reaching statistical significance. Patients in both groups had associated dyspeptic symptoms (Table 1).

### 3.2. Esophagogastric Junction Morphology

A type 2 or type 3 EGJ morphology was observed in 20 patients (64.5%) from the IEM-4 group and 13 (81.2%) patients from the borderline group, but the difference was not statistically significant (Z = −1.2, *p* = 0.23) (Table 1).

We compared manometry parameters in the IEM-4 vs. borderline group. In the IEM-4, the mean number of swallows that were analyzed per patient was 9.6 (min–max: 7–11), while, in the borderline group, the number was 9.8 (min–max: 8–10). There was no difference between the resting LES pressure, the mean IRP, the esophageal length, or EGJ length between the two groups. As expected, patients from the IEM-4 group had more ineffective swallows in total (*p* = 0.001) and more failed swallows (*p* = 0.002) compared to the borderline group (Table 2).

We compared the number of ineffective swallows in patients with or without reflux symptoms and the difference was not significant (7.2 vs. 8.4, *p* = 0.13). Similarly, type 2 or 3 EGJ morphology was not significantly more often in patients with reflux symptoms (chi-square test = 0.72) compared to those without reflux symptoms. We could not identify a cut-off value for the LES resting pressure below which reflux symptoms were more common (Table 3).

## 4. Discussion

Our study showed that using the Chicago v4.0 criteria, the prevalence of IEM decreased from 30.1% to 21.7%, and 11.8% of patients had an inconclusive diagnosis of IEM. Dysphagia, but not chest pain, was more common in the group of patients with IEM based on Chicago v4.0. Patients from the borderline group more often had type 2 or 3 EGJ morphology and reflux symptoms but without reaching statistical significance.

IEM was part of minor motility disorders in the third iteration of esophageal motility disorders. In asymptomatic subjects, around 7% have manometric criteria for IEM [16]. The change in the diagnosis criteria of IEM came from observations that only severe IEM, characterized by more than 70% of ineffective swallows, was associated with abnormal esophageal acid burden [17]. Other studies also reported that severe IEM is associated with pathological acid exposure [18,19]. An impaired bolus clearance seems to be the main contributor to this increased acid burden. Recent studies combined manometry and esophageal impedance and showed that ≥30% failed contractions or ≥70% ineffective swallows predicted an altered bolus clearance. The absence of contraction reserve did not predict the bolus clearance [16].

The prevalence of IEM varied in previous studies. Using Chicago v3.0 criteria, Chug et al. reported that 13.7% of their patients had IEM [9]. A large recent study [20] reported in patients without obstruction of the EGJ or hypermotility a prevalence of borderline IEM (patients with 50–70% of ineffective swallows, considered as mild IEM) in 21.7% of patients and definite IEM based on Chicago v4.0 (or severe IEM) in 32.4% of patients. Our results showed significantly fewer patients with a definite diagnosis of IEM based on Chicago v4.0 criteria.

When it comes to the relationship between the type of ineffective swallows and the associated symptoms, the existing data provide inconclusive information. In one study, it was noted that patients with IEM did not exhibit differences in the severity of symptoms such as dysphagia, chest pain, regurgitation, or weight loss compared to those with normal motility [9]. Another study reported that abnormal esophageal acid exposure was present in over half of the patients but, regardless of acid exposure, these patients had a similar prevalence of symptoms, including dysphagia and reflux symptoms [21]. Another study failed to find any correlation between manometric changes (including contraction pattern) and symptoms [22]. In patients with IEM, particularly those with a very low DCI (<100 mmHg·s·cm), there was a noticeable reduction in bolus clearance and more pronounced dysphagia. This emphasized the significance of DCI as the most functionally relevant metric when evaluating IEM over peristaltic reserve [23]. On the other hand, Roman et al. found that frequent breaks in the 20 mm Hg isobaric contour were correlated with unexplained nonobstructive dysphagia [13]. Failed peristalsis was not more frequent in dysphagic patients, as one would have expected. At that time, swallows with breaks were classified into small and large breaks and it is possible that some of the swallows with small breaks were hypotensive. The DCI parameter was not used in this study. Our study showed that most patients in the IEM-4 group had dysphagia and reflux symptoms. Patients with inconclusive diagnosis experienced the same symptoms, with dysphagia being less prevalent, suggesting that dysphagia is reported more often by patients with severely altered esophageal motility. In fact, over time, the criteria for IEM changed several times, the first threshold for ineffective swallows being 30%. The reason for changing the criteria was that higher thresholds correlated better with dysphagia and heartburn [4].

In their study, Dao et al. [20] compared patients with IEM having more than 70% ineffective swallows (severe IEM) with the borderline cases (50 to 70 inefficient swallows- mild IEM). Severe IEM and absent contractility, but not mild IEM, were significantly associated with Los Angeles (LA) grade B–D esophagitis. This study suggested that patients with severe IEM are at a higher risk for developing more serious forms of esophagitis compared to those with milder forms of IEM [20]. None of our patients referred to manometry had Los Angeles B, C, D, or Barret’s esophagus. However, among patients with normal esophageal mucosa, we had both patients with severe IEM and borderline criteria for IEM. We did not include patients diagnosed with absent contractility in this analysis. Similarly, Pakoz et al. [24] showed that both patients with pathological and physiological reflux have IEM, based on Chicago v3.0 criteria. In their study, esophagitis rates were higher in the pathological reflux group (observed in almost half of the patients), but LES resting pressure, IRP, the presence of hernia, and the rate of severe IEM were similar between the two groups.

Ineffective esophageal motility is a common motor dysfunction observed in patients with GERD, even in the absence of major pathological changes. A poor peristalsis was associated with delayed distal esophageal acid clearance and more reflux during supine position. However, it is not clear whether IEM is reversible after GERD treatment [6], even if there are some data that IEM improved after Nissen fundoplication [25]. Smooth muscle relaxants can also influence esophageal peristalsis. However, when compared to controls, IEM patients were not more often prescribed drugs that can interfere with the peristaltic function. In patients with IEM, those with a lower mean DCI and more ineffective swallows were significantly more likely to be using skeletal muscle relaxant medications (Cyclobenzaprine, Tizanidine, Methocarbamol, Metaxalone, and Baclofen) [26]. IEM was more often reported in patients with systemic sclerosis [7], diabetes mellitus [8], and eosinophilic esophagitis [27]. Other disorders associated with IEM are amyloidosis, acute ethanol ingestion, chronic alcoholism with neuropathy, adenocarcinoma, or endoscopic submucosal dissection [4,28]. Studies using high-frequency endoscopic ultrasound showed that the esophageal wall is significantly thicker in patients with IEM not related with GERD, compared to patients with IEM and GERD, suggesting different pathophysiologic mechanisms for IEM: a possible primary esophageal muscular disorder in the first case and a manometric abnormality induced by chronic acid-reflux exposure in the later [29].

Our study has several limitations. Our center is a referral center for dysphagia and achalasia, and almost half of our procedures were performed on patients with obstruction of the EGJ, prior to or after intervention. This might explain the higher prevalence of achalasia in our study but, at the same time, the lower prevalence for other esophageal motility disorders. Another limitation comes from the fact that only patients with no changes on upper GI endoscopy that could explain their symptoms were referred to manometry. Therefore, we did not have patients with Los Angeles B–D esophagitis in our cohort. We cannot say whether IEM is more often observed in this group of patients, but we can state based on this research that the prevalence of IEM based on Chicago 4.0 is high in symptomatic patients with normal upper GI endoscopy. Unfortunately, only a minority of patients with reflux symptoms had ambulatory pH impedance (to rule in or rule out GERD), and the data were not presented.

The more stringent criteria proposed in the Chicago v4.0 for esophageal motility disorders, including IEM, were meant to reduce ambiguity by more standardized and rigorous criteria and, at the same time, to determine the patterns that are clinically relevant. There are data that IEM with >70% of ineffective swallows is accompanied by impaired bolus transit and is more likely to have a high acid burden [18,19]. Our research showed that severe IEM is also observed in patients with normal upper GI endoscopy, which are more likely to have normal esophageal acid exposure, suggesting that acid burden is relevant only in a subset of patients. This idea is supported by the observation that patients with IEM and physiological reflux were younger compared to those with IEM and pathological reflux [24]. IEM might develop earlier, in the absence of reflux. We did not find other relevant differences between IEM-4 patients and borderline patients. Future research should also focus on this category of patients with IEM that is not related to GERD.

Manometry is recommended before anti-reflux surgery to exclude severe motility disorders like achalasia. Altered esophageal peristalsis, including a diagnosis of IEM, was reported in 29.1% of patients. However, a change in the surgical procedure from a complete to a partial fundoplication was reported only in 17.3% of patients. The latest consensus regarding the treatment of GERD stated that manometry should be used for all patients with refractory reflux symptoms undergoing preoperative evaluation [30]. The guideline also recommended an adjusted surgical technique for GERD, based on the presence or absence of esophageal dysmotility, but admitted that there is limited evidence to support such a recommendation [30]. There are data in the literature that manometry cannot predict postoperative dysphagia, and some specialists argue that fundoplication is safe in patients with IEM [31]. The use of provocative tests such as MRS, as recommended in Chicago v4.0, could be useful in this situation. One study reported that the absence of a strong contraction at the end of MRS sequence predicts late postoperative dysphagia after anti-reflux surgery [32]. The impact of these stringent criteria for IEM diagnosis in the evaluation prior to anti-reflux surgery remains to be established. There is also interest in the intraoperative diagnosis of esophageal motility disorder. One study compared preoperative manometry with intraoperative impedance planimetry and panometry (endo-FLIP) to assess esophageal peristalsis. The technique was highly sensitive to identify peristaltic dysfunction [33].

## 5. Conclusions

In our cohort of patients referred for manometry, using the Chicago v4.0 criteria, only 8.9% of patients had a definite diagnosis of IEM. Among patients without esophagogastric junction outflow obstruction, 1 in 5 patients had IEM, while 1 in 10 was inconclusive for a diagnosis of IEM. Another important finding was that dysphagia was the only symptom more common in patients, classified as IEM based on Chicago v4.0, compared to patients with borderline IEM criteria. Therefore, we might consider dysphagia as the relevant symptom in patients with IEM. Other symptoms, including reflux symptoms, were similar. Among manometry parameters, as expected, only DCI and the number of ineffective swallows were significantly different in the IEM-4 group. The morphology of the EGJ did not differ between the IEM-4 and borderline group nor did LES resting pressure or IRP. Studies should also focus on patients with IEM and normal esophageal acid exposure, as a subset of patients with IEM.

## Figures and Tables

**Figure 1 medicina-60-01469-f001:**
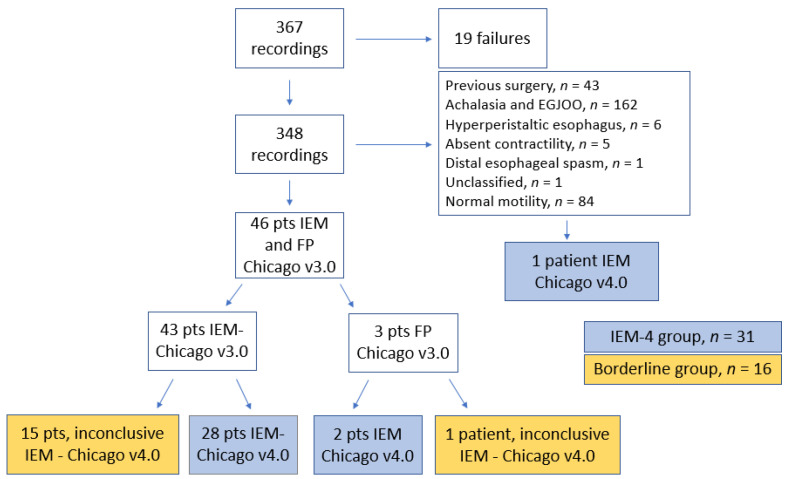
Selection of patients with ineffective esophageal motility (IEM) and fragmented peristalsis (FP) according to Chicago v3.0 and Chicago v4.0 criteria.

**Table 1 medicina-60-01469-t001:** Demographic and clinical data of patients with IEM based on Chicago v4.0 (IEM-4 group) compared to patients with an inconclusive diagnosis for IEM (borderline group).

	IEM-4 (*n* = 31)	Borderline (*n* = 16)	*p*
Age, mean ± SD	50 ± 16.6	52.7 ± 12.6	0.64
Gender, females, *n* (%)	14 (45.1)	11 (68.7)	0.12
BMI, kg/m^2^	25.9 ± 3.4	25.2 ± 4.3	0.54
Lean *n* (%)	12 (40)	6 (42.8)	0.94
Reflux symptoms *n* (%)	22 (70.9)	14 (87.5)	0.20
Dysphagia *n* (%)	24 (77)	7 (44)	0.02 (Z = 2.3)
Chest pain *n* (%)	16 (51.6)	6 (37.5)	0.36
Cough *n* (%)	9 (29.0)	5 (31.3)	0.87
Belching *n* (%)	16 (51.6)	11 (68.8)	0.26
Nausea *n* (%)	8 (25.8)	7 (43.8)	0.21
Bloating *n* (%)	14 (45.2)	9 (56.3)	0.47
Postprandial fullness *n* (%)	3 (9.7)	0 (0)	0.19
Early satiation *n* (%)	6 (19.4)	1 (6.3)	0.23
Globus *n* (%)	5 (16.1)	4 (25)	0.47
Weight loss *n* (%)	5 (16.1)	0 (0)	0.09
Hiccups *n* (%)	1 (3.2)	2 (12.5)	0.22
ENT symptoms *n* (%)	1 (3.2)	1 (6.3)	0.62
Median dysphagia duration, months (median, IQ range)	12 (IQ: 0.25−12)	0 (IQ: 0−9)	0.24
**EGJ morphology**			
EGJ morphology type 2 or 3, *n* (%)	20 (64.5)	13 (81.2)	0.23
EGJ type 3—Hiatal hernia	5 (3 to 6 cm)	5 (3 to 5 cm)	

BMI, body mass index; IEM-4, patients with ineffective esophageal motility based on Chicago v4.0 classification; Borderline group, patients with ineffective esophageal motility or fragmented peristalsis based on Chicago v3.0 classification but with an inconclusive diagnosis using Chicago v4.0 criteria; EGJ, esophagogastric junction.

**Table 2 medicina-60-01469-t002:** Manometric parameters in IEM-4 group compared to borderline group.

Parameter	IEM-4, *n* = 31	Borderline, *n* = 16	*p*
Resting LES pressure (mmHg), mean ± SD	30.4 ± 11.5	32.7 ± 10.9	0.22
IRP (mmHg), mean ± SD	14.4 ± 6.5	12.9 ± 6.9	0.84
Esophageal length, cm (median, IQ range)	22 (IQ: 21–23)	22 ± 2.7	NS
EGJ length, cm (median, IQ range)	3 (IQ: 2–3.9)	3.5 (IQ: 3–4.5)	0.13
DCI mmHg·s·cm (median, IQ range)	258 (IQ: 170–473)	462 (IQ: 375–718)	0.001
Ineffective swallows, (median, IQ range)	9 (IQ: 7–10)	6 (IQ: 5–6)	0.001
Failed peristalsis, mean ± SD	5.2 ± 2.5	2.9 ± 1.7	0.002
Hypotensive swallows, (median, IQ range)	2 (IQ: 1–3)	2 (IQ: 1–2.5)	0.72
Fragmented swallows, (median, IQ range, min–max)	0 (IQ: 0–1; range: 0–7)	0 (IQ: 0–0; range: 0–6)	0.08

IEM-4, patients with ineffective esophageal motility based on Chicago v4.0 classification; Borderline group, patients with ineffective esophageal motility or fragmented peristalsis based on Chicago v3.0 classification but with an inconclusive diagnosis using Chicago v4.0 criteria; IRP, integrated relation pressure; IQ range, interquartile range; DCI, distal contractile integral.

**Table 3 medicina-60-01469-t003:** Distribution of patients based on EGJ morphology, LES resting pressure, and presence or absence of reflux symptoms. For different values of LES resting pressure, we assessed the number of patients with or without reflux symptoms.

EGJ Morphology	Reflux Symptoms	No Reflux	Chi-Square Test
Type 2 and 3, *n* = 32	25	7	0.72
Type 1, *n* = 15	11	4	
LES resting pressure, mmHg			
≤20	4	1	0.85
>20	32	10	
≤24	10	1	0.2
>24	26	10	
≤26	16	2	0.12
>26	20	9	
≤28	17	3	0.24
>28	19	8	
≤30	19	4	0.35
>30	17	7	
≤35	23	7	0.98
>35	13	4	

## Data Availability

The data are not publicly available due to privacy and ethical restrictions imposed by the public hospital, in accordance with institutional policies and applicable regulations.

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
