# Peer review of "Comparative Prevalence of Ineffective Esophageal Motility: Impact of Chicago v4.0 vs. v3.0 Criteria"

_medicina, 2024, doi:10.3390/medicina60091469_

Round 1

Reviewer 1 Report

Comments and Suggestions for Authors

Reviewer Comments:

Congratulations to the authors on this interesting manuscript. Surdea-Blaga et al. evaluated the manometric findings of ineffective esophageal motility (IEM). The study’s topic is relevant and provides valuable insights into esophageal motility disorders. However, I have some remarks to enhance the manuscript's clarity and precision.

Study Objective:

  1. The last paragraph of the introduction is somewhat confusing. To improve clarity, I suggest subdividing the objectives into primary and secondary.
  2. Additionally, consider modifying the title to align with the primary study objective. For example, if the main objective was to evaluate the change in IEM prevalence, this should be reflected in the title.

Methods:

  1. Clarify the study design. Instead of just stating that it is retrospective, provide more details on the study design.
  2. Specify the statistical strategy used to determine if variables assumed parametric or nonparametric distribution. This will provide better insight into the data analysis process.

Writing:

  1. Correct the terminology from "Barret esophagus" to "Barrett's esophagus."
  2. Use superscript formatting for square meters in kg/m².
  3. Format "vs." in italics.
  4. Use the symbol "≤" instead of "<=" for consistency and accuracy.

Author Response

Responses to Reviewer 1

We greatly appreciate your thoughtful comments and feedback on our manuscript. Your constructive feedback and suggestions have significantly enhanced the clarity, precision, and overall quality of our work. We have carefully considered and incorporated all your comments, and we are grateful for your valuable insights.  Below, we have provided detailed responses to each of your comments. Thank you again for your valuable insights.

Reviewer 1

Congratulations to the authors on this interesting manuscript. Surdea-Blaga et al. evaluated the manometric findings of ineffective esophageal motility (IEM). The study’s topic is relevant and provides valuable insights into esophageal motility disorders. However, I have some remarks to enhance the manuscript's clarity and precision.

Study Objective:

  1. The last paragraph of the introduction is somewhat confusing. To improve clarity, I suggest subdividing the objectives into primary and secondary.

RESPONSE – We rephrased the objectives to make it clearer.

“The primary aim of this study was to determine how the Chicago v4.0 criteria changed the IEM prevalence in a cohort of patients referred to esophageal manometry. The secondary aim was to describe the clinical and manometric characteristics of patients with IEM based on Chicago v4.0 compared to those that now fall in the category of “inconclusive diagnosis of IEM”.”

  1. Additionally, consider modifying the title to align with the primary study objective. For example, if the main objective was to evaluate the change in IEM prevalence, this should be reflected in the title.

We changed the title Ineffective Esophageal Motility: Clinical and Manometric Characterization with a new one, that we hope is more related with our paper.

“Comparative Prevalence of Ineffective Esophageal Motility: Impact of Chicago v4.0 vs. v3.0 Criteria”

Methods:

  1. Clarify the study design. Instead of just stating that it is retrospective, provide more details on the study design.

RESPONSE - As suggested, we introduced in the section Material and Methods a new paragraph regarding study design.

2.1. Study design

From our database of esophageal HRM recordings (performed from November 2015 to April 2023) we selected all patients with a diagnosis of IEM or FP, based on Chicago v3.0 criteria. All identified tracings were reanalyzed and reclasified using the Chicago v4.0 criteria, to see how the prevalence of IEM changed using the new criteria. Some of the patients were borderline based on the new clasification system (“inconclusive for a diagnosis of IEM”). For the secondary objective of the study, we compared the clinical and manometric parameters between patients were still clasified as IEM based on Chicago v4.0 and the patients with an inconclusive diagnosis for IEM, using Chicago v4.0 criteria.

We changed accordingly other paragraphs in the section reffering to Methods

The study was retrospective. All tracings of patients diagnosed with IEM or FP based on Chicago v3.0, irrespective of their symptoms, were selected from our database of esophageal HRM recordings……

All tracings classified as IEM or PF using Chicago v3.0 were reanalyzed using the Chicago v4.0 criteria. A diagnosis of IEM was based either on the presence of more than 70% ineffective swallows (including weak, failed, or fragmented swallows) or at least 50% failed swallows. Tracings with 50% to 70% ineffective swallows, were considered inconclusive for IEM, based on Chicago v4.0 criteria (1).

  1. Specify the statistical strategy used to determine if variables assumed parametric or nonparametric distribution. This will provide better insight into the data analysis process.

We added in statistical analysis a phrase regarding the evaluation of data distribution:

To assess whether the variables followed a parametric or nonparametric distribution, we evaluated skewness and kurtosis values. Variables with skewness or kurtosis outside the range of -1 to +1 were considered non-normally distributed and were reported as median and interquartile range (25th, 75th percentiles).

Writing:

  1. Correct the terminology from "Barret esophagus" to "Barrett's esophagus."
  2. Use superscript formatting for square meters in kg/m².
  3. Format "vs." in italics.
  4. Use the symbol "≤" instead of "<=" for consistency and accuracy.

Thank you for pointing out the writing errors. We have carefully reviewed the manuscript and corrected all identified mistakes. We appreciate your attention to detail, which has helped improve the clarity and accuracy of our paper.

Reviewer 2 Report

Comments and Suggestions for Authors

The aim of the article is to review the threshold of ineffective esophageal motility using new criteria, taking into account the differences of all types presented by patients. Unfortunately, the study is retrospective. The work is based, rightly, a lot on esophageal manometry induced by symptoms, and on measurements in mm of Hg that ranged from 100, failed contraction, present in collagen diseases, diabetes mellitus in esophageal reflux, to 450, ineffective contraction. Obviously, dynamometric changes must always be accompanied by symptoms. A large number of patients with inconsistent swallowing were recruited for the study. The working method is perfectly described so that it is absolutely reproducible. With their study, the research colleagues have demonstrated that the failure of esophageal motility has decreased from 30 to 21% according to the Chicago criteria. In the discussion the results obtained are widely discussed and find a fair acceptable explanation. On the other hand we must also consider the purely surgical aspect of the matter, in fact manometry should also be used in the operating room when making the anti-reflux plastic. In the article this aspect is not considered but if we count how many interventions of this type are performed daily with often empirical measurements, we should take certain studies into greater consideration. Good iconography, English to be revised, the bibliography is a good basis for the proposed study

Comments on the Quality of English Language

english nedds to be revised

Author Response

Responses to Reviewer 2

We appreciate your useful comments and feedback on our manuscript. Your constructive feedback and recommendations have significantly enhanced the clarity, precision, and overall quality of our work. We have carefully considered and incorporated all your comments, and we are grateful for your valuable insights.  Below, we have provided detailed responses to each of your comments. Thank you again for your valuable insights.

Reviewer 2

The aim of the article is to review the threshold of ineffective esophageal motility using new criteria, taking into account the differences of all types presented by patients. Unfortunately, the study is retrospective. The work is based, rightly, a lot on esophageal manometry induced by symptoms, and on measurements in mm of Hg that ranged from 100, failed contraction, present in collagen diseases, diabetes mellitus in esophageal reflux, to 450, ineffective contraction. Obviously, dynamometric changes must always be accompanied by symptoms. A large number of patients with inconsistent swallowing were recruited for the study. The working method is perfectly described so that it is absolutely reproducible. With their study, the research colleagues have demonstrated that the failure of esophageal motility has decreased from 30 to 21% according to the Chicago criteria. In the discussion the results obtained are widely discussed and find a fair acceptable explanation. On the other hand we must also consider the purely surgical aspect of the matter, in fact manometry should also be used in the operating room when making the anti-reflux plastic. In the article this aspect is not considered but if we count how many interventions of this type are performed daily with often empirical measurements, we should take certain studies into greater consideration. Good iconography, English to be revised, the bibliography is a good basis for the proposed study

Response to Reviewer 2

Thank you for your thorough and insightful review of our manuscript. We appreciate your positive feedback regarding our study's methodology, discussion, and iconography.

Regarding your suggestion to consider the surgical aspect, particularly the use of manometry in the operating room during anti-reflux procedures, we agree that this is an important consideration. While our study primarily focuses on the diagnostic and evaluative aspects of esophageal motility using the new Chicago criteria, we recognize the relevance of manometric data in surgical decision-making, particularly in anti-reflux surgery.

We revised the discussion section to briefly acknowledge the importance of manometry in the surgical context and suggest it as an area for future research. We added this paragraph in discussions

 Manometry is recommended before anti-reflux surgery to exclude severe motility disorders like achalasia. Altered esophageal peristalsis, including a diagnosis of IEM was reported in 29.1% of patients, but a change in the surgical procedure, from a complete to a partial fundoplication was reported only in 17.3% of patients. The latest consensus regarding the treatment of GERD, stated that manometry should be used for all patients with refractory reflux symptoms undergoing preoperative evaluation (30). The guideline also recommended adjusted surgical technique for GERD, based on the presence or absence of esophageal dysmotility, but admitted that there is limited evidence to support such a recommendation (30). There is data in literature that manometry cannot predict post-operative dysphagia, and some specialists argue that fundoplication is safe in patients with IEM (31). The use of provocative tests as recommended in Chicago v4.0, could be useful in this situation. One study reported that the absence of a strong contraction at the end of this sequence predicts late postoperative dysphagia after anti-reflux surgery (32). The impact of these stringent criteria for IEM diagnosis in the evaluation prior to anti-reflux surgery remains to be established.

To our knowledge, at least one study compared pre-operative manometry with intraoperative endo-FLIP to assess esophageal peristalsis. [VanDruff VN, Amundson JR, Joseph S, Che S, Kuchta K, Zimmermann CJ, Ishii S, Hedberg HM, Ujiki MB. Impedance planimetry and panometry (EndoFLIP™) can replace manometry in preoperative anti-reflux surgery assessment. Surg Endosc. 2024 Jan;38(1):339-347.], and there is a lot of potential for future research in this area. 

This paragraph was added at the end of discussions

There is also interest in the intra-operative diagnosis of esophageal motility disorder. One study compared pre-operative manometry with intraoperative Impedance planimetry and panometry (endo-FLIP) to assess esophageal peristalsis. The technique was highly sensitive to identify peristaltic dysfunction (32).

We also appreciate your note about the English language and will ensure a thorough revision of the manuscript to improve clarity and correctness. Thank you again for your valuable feedback, which has helped us to enhance the quality of our work.

---------------------------------------------------------------------------------------------------------------------

In addition, we expanded one of the paragraphs from introduction

Patients presenting with 50 to 70% ineffective swallows during esophageal manometry are considered borderline, and the manometric findings are deemed “inconclusive for a diagnosis of IEM”. In such cases, further diagnostic evaluations are warranted to clarify the diagnosis. Additional testing may include combined impedance/HREM  (9,10), which can provide insights into bolus transit and the correlation with symptoms. An impaired bolus clearance detected using impedance was directly associated with the percentage of ineffective swallows recorded during manometry. In addition, patients with IEM showed a marked decrease in impedance values, indicating poor esophageal clearance  (9,10). Barium esophagogram can visually assess bolus passage through the esophagus and identify areas of stasis, impaired transit or delayed esophageal emptying (11). Moreover, performing multiple rapid swallows (MRS) during high-resolution manometry can evaluate the esophageal contractile reserve; a lack of contraction reserve on MRS may support the diagnosis of IEM when the percentage of ineffective swallows is borderline (12).
